# IMPON: Efficient IMPortance sampling with ONline regression for rapid neural network training

**Vignesh Ganapathiraman, Francisco Calderon Rodriguez, Anila Joshi**
Amazon
vignesh117@gmail.com, {fccal, anilajo}@amazon.com

## Abstract

Modern-day deep learning models are trained efficiently at scale thanks to the widespread use of stochastic optimizers such as SGD and ADAM. These optimizers update the model weights iteratively based on a batch of uniformly sampled training data at each iteration. However, it has been previously observed that the training performance and overall generalization ability of the model can be significantly improved by selectively sampling training data based on an importance criteria, known as *importance sampling*. Previous approaches to importance sampling use metrics such as loss, gradient norm etc. to calculate the importance scores. These methods either attempt to directly compute these metric, resulting in increased training time, or aim to approximate these metrics using an analytical proxy, which typically have inferior training performance. In this work, we propose a new sampling strategy called ***IMPON***, which computes importance scores based on an auxiliary linear model that regresses the loss of the original deep model, given the current training context, with minimal additional computational cost. Experimental results show that IMPON is able to achieve a significantly high test accuracy, much faster than prior approaches.

## 1  Introduction

Modern machine learning systems are a complex network of sub-components such as data ingestion, architecture search, hyper parameter optimization (HPO), model compression etc. Apart from data ingestion component, all other components need some sort of model evaluation and selection. Model selection typically involves running several iterations of training, often until full convergence, which directly translates to increase in overall time taken to obtain an optimal model. This effect is only exacerbated with increase in the size of the data and model. However, model selection often does not require training until full convergence. For instance, in HPO we only compare the training performance of several configurations of the same model. Therefore, if we can achieve highly accurate models much earlier in the training, we can avoid training the model until full convergence and subsequently reduce the model selection time. In this work we aim to achieve the same.

Deep learning models are often trained using stochastic "gradient-based" learning algorithms such as SGD and Adam Kingma & Ba (2014). The convergence of these algorithms are often stymied due to the uniform sampling strategy of the training data employed by these methods Katharopoulos & Fleuret (2018, 2017). Recently, researchers have looked at importance sampling for directly addressing this issue. Importance sampling chooses training samples that are important to the prediction task at hand based on a predefined importance metric, so that examples that really matter are learned by the model earlier in the training, resulting in reduced training times. However, computing importance scores often incurs additional computational costs and their complexity can vary based on the importance criteria.

Has it Trained Yet? Workshop at the Conference on Neural Information Processing Systems (NeurIPS 2022).

Existing works on importance sampling form a broad spectrum, in terms of their computational complexity. On the one hand, there are methods Katharopoulos & Fleuret (2017); Jiang et al. (2019) that use metrics such as training loss or gradient norms as importance scores, which are generally expensive to compute, but are found to be good indicators of sample importance. In contrast, some works have used other metrics such a model uncertainty Katharopoulos & Fleuret (2018) and other proxies for training loss Chang et al. (2017), which are cheaper to compute but have been shown to be not as effective.

In this work, we propose IMPON, a smart sampling technique that uses training loss as the importance score. IMPON uses a simple online linear model to estimate the expected loss of a small bank of training samples at each iteration. Important samples for a training batch are then chosen from this data bank directly. In our experiments we see that IMPON is able to obtain higher validation accuracies, significantly faster, with very minimal computational overhead.

## 2   Related works

Recently the idea of using techniques based on importance sampling, for improving convergence rates of stochastic optimization algorithms, have received a lot of attention. Prior work such as Alain et al. (2015); Loshchilov & Hutter (2015); Schaul et al. (2015); Bengio et al. (2009); Katharopoulos & Fleuret (2018, 2017) have tried importance sampling for deep learning objectives. Alain et al. (2015); Katharopoulos & Fleuret (2018) have proven that choosing examples proportional to the gradient norm can enable faster convergence. However evaluating the gradient of each training example at every iteration is computationally expensive.

Few other works like Katharopoulos & Fleuret (2017); Schaul et al. (2015); Loshchilov & Hutter (2015) have used loss values as the importance scores. Despite the fact that training losses don't directly correspond to high gradients, loss values are much cheaper to compute than gradient norm, and are a reasonable proxy for gradient norms. Having said that loss computation still requires a full forward pass over the entire network. To address this issue, Schaul et al. (2015); Loshchilov & Hutter (2015) maintain the history of input examples and its corresponding loss values and sample examples based on its historical loss values. Katharopoulos & Fleuret (2017) has trained a LSTM model with input examples and loss values to estimate the importance scores (losses). In a similar realm, Jiang et al. (2019) propose the selective backpropagation method, where loss values are computed for all the training examples, but backward pass is only computed for the important samples.

Few other works have aimed to predict importance scores with a computationally efficient strategy. For instance, Chang et al. (2017) proposed a sampling strategy called *P-SGD* where he predicted importance scores based on the average predicted probability across all previous epochs associated to the ground truth of each sample: $p_i^t = \frac{1}{t} \sum_{k=1}^{t} \left(1 - y_i^T h_\theta^k(y|x_i)\right) + \epsilon^t$, where $p_i$ is the importance score of training example $i$, $t$ is the current epoch, $h_\theta^k(y|x_i)$ is the prediction of the model for the input example $x_i$ at epoch $k$, and $\epsilon^t$ is a smoothness constant. This approach avoids the extra forward passes that is required in computing the importance score using loss values and can be computationally efficient. Another similar approach Chang et al. (2017) selects samples that are closer to the decision boundaries and chooses samples with higher uncertainty. They define importance as $c_i^t = p_i^t \times (1 - p_i^t)$. However in hindsight, although computationally efficient, these methods don't aggressively sample examples that improve the model earlier on in the training. As a result, we observe that these methods converge much slower than the previous methods.

In this work, we aim to reduce computational complexity while using loss values to estimate importance scores, similar to Katharopoulos & Fleuret (2017, 2018). We avoid a full forward pass to obtain loss values by estimating the loss values just like Katharopoulos & Fleuret (2017). However, we train a simple linear online model instead of large sequence models as in Katharopoulos & Fleuret (2017) to estimate the loss based on the current training context, thereby reducing the per-iteration cost and overall training time.

## 3   Method

The overall training pipeline using the IMPON sampler is illustrated in Figure 1. IMPON has 3 major components, which we detail below.

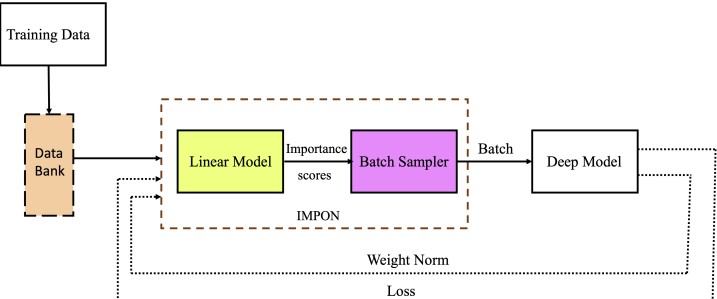

Figure 1: The IMPON training pipeline. The IMPON sampler consists of a linear model that predicts importance scores and a batch sampler.

- **Data bank.** A small memory bank, that holds a randomly sampled pool of examples on which importance scores are computed. The size of the data bank $N$ is typically much smaller than the entire training data (in our experiments, we use a bank size of 200).

- **Linear model.** This is the most important component of IMPON. The linear model estimates the expected loss of each sample in the data bank. The input to the linear model is the *context vector* $c_{x_i}$, which is described as $c_{x_i} = \{x_i; ||w_i||_F\}$, where $x_i$ is a feature representation of an example from the data bank in iteration $i$ and $w_i$ is the current weight matrix of the deep model. The norm of $w_i$, serves as a snapshot of the state of the deep model. The IMPON model is trained by an SGD optimizer using the actual losses from the deep model after performing a forward pass.

- **Batch sampler.** The Batch sampler, chooses $b$ important samples, from the $N$ samples in the data bank, based on the importance scores (expected loss) computed by the linear model. These $b$ samples are provided as the training batch for training the deep model in iteration $i + 1$.

The training pipeline consists of the following steps:

1. At iteration $i$, randomly sample $N$ samples from the training data to the data bank.
2. Construct the context vector $c_{x_i}$ for samples in the data bank.
3. Predict the importance scores for all the samples in the data bank using the linear model.
4. Select $b$ samples with the highest importance scores from the data bank and use that to perform a single training iteration of the deep model (forward + backward pass) and observe the training loss $\ell_{x_i}, i \in \{1, \ldots, b\}$.
5. Use $\{c_{x_i}, \ell_{x_i}\}$ to update the linear model in an online fashion.

**Weight reset.** Note that the linear model is updated by regressing against the actual training loss in each iteration. In the earlier stages of training, this loss can be high and also quite noisy and as a result the loss estimates learned by the online model can also be unreliable. In our experiments, we observed that the linear model overfitted to these noisy losses in the first few epochs, and resulted in the deep model converging with high loss too early. To alleviate this, we propose to reset the weights of the linear model after every epoch of the deep model training. We note here that alternative strategies for manipulating the weights of the linear model may be considered and might be more effective. However, we observed that just resetting the weights resulted in a far more stable training.

## 4 Experiments

**Dataset & Training details.** To show the effectiveness of our proposed approach, we trained a ResNet18 model on four image classification datasets: 1) SVHN Netzer et al. (2011) 2) CIFAR10 Krizhevsky et al. (2009) and 3) CIFAR100 Krizhevsky et al. (2009).

We followed the experimental setup of Arazo et al. (2021) and trained the ResNet18 model for about 30 iterations on the above mentioned datasets and observed the training loss in every iteration and validation accuracies after every epoch. The learning rate for all the baseline models were set to $1e^{-1}$ for all the datasets, with a momentum of $0.9$ for SGD-based optimizers. For IMPON, we are

dealing with two separate optimizers, one for training the deep model and another for the IMPON linear model. We observed that setting a high learning rate for training the deep model resulted in overfitting of the downstream online model. So, we used learning rates of $1e^{-3}$ and $1e^{-6}$ for the deep model and the linear model respectively.

**Baselines.** We compared IMPON with other optimizers that fall in one of the 3 categories: a) Optimizers that do uniform sampling (vanilla SGD and Adam). b) Optimizers that do importance sampling but incur a relatively "high" per-iteration sampling cost (selective backpropagation Jiang et al. (2019)). c) Optimizers that do importance sampling but are computationally efficient (p-SGD Katharopoulos & Fleuret (2018), c-SGD Chang et al. (2017). Additionally, we also included a version of IMPON where we chose the examples with the least importance score as predicted by the linear model in each iteration. We call this method *IMPONwLowScores*.

**Results.** The results on the image classification task are given in 3 and 2. IMPON is able to outperform all the baselines across all the datasets comprehensively. The effect of our aggressive importance sampling is especially pronounced in the first few epochs, where IMPON is able to achieve close to $30\%$ higher validation accuracy compared to the nearest competitor (Adam on SVHN).

We like to point out that all the SGD-based optimizers eventually converge to similar validation accuracies and training loss at the end of the training. This makes sense, as none of the importance sampling methods modify the underlying optimizer (SGD), but tweak the sampling process. However, IMPON excels at achieving a high accuracy much earlier on in the training.

**IMPON hyperparameters.** The IMPON block as shown in 1 has 2 parameters that can be tuned. 1) Learning rate of the optimizer for the IMPON linear model 2) Size of the data bank. Since the IMPON linear model is trained jointly with the ResNet18 model, a high learning rate of the linear model (especially earlier on in the learning cycle) resulted in overfitting. A similar effect is observed with the size of the data bank, which is discussed in detail in Appendix 5.

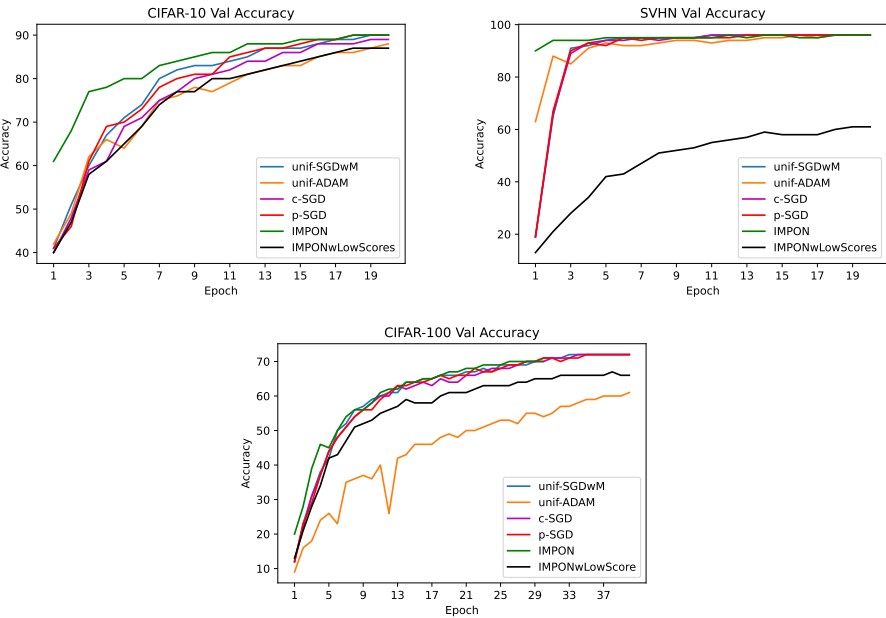

Figure 2: Validation accuracies on popular image classification datasets. IMPON reaches a high validation accuracy in the first few training epochs compared to the baselines.

# 5 Ablation on the size of the data bank

The size of the data bank had a significant impact on the overall model performance. In general, we observed that a bigger data bank negatively impacted the overall model performance, as shown in 4.

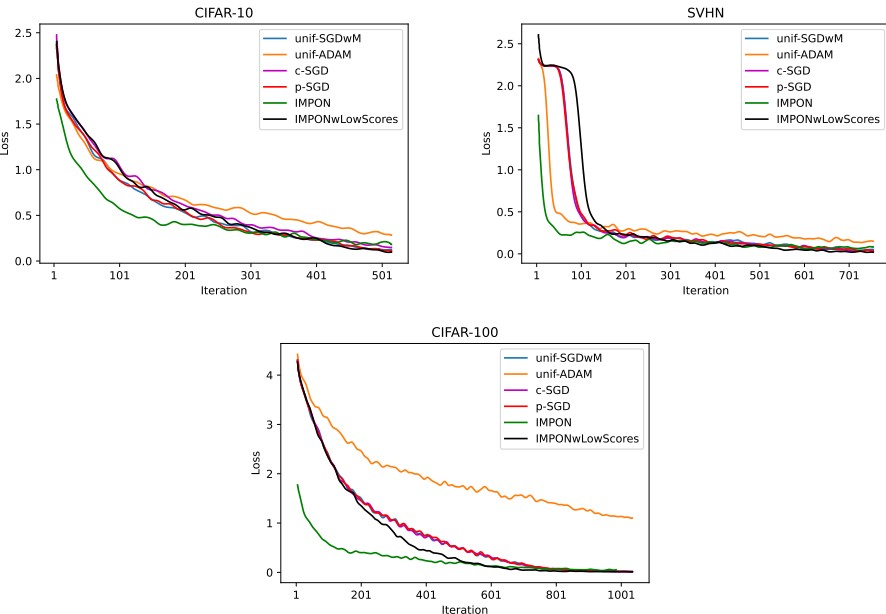

Figure 3: Training losses on popular image classification datasets. IMPON attains a much lower training loss in the first few training iterations compared to the baselines.

Particularly we saw that the model performance gets high in the first few epochs and then plateaus. We hypothesize that with a higher data bank, the IMPON online model is able to confidently assign importance scores to a large portion of the training data. In the earlier stages of training, such a strategy could be detrimental as the training signals (loss values in our case) can be noisy in the earlier iterations. This effect is ameliorated somewhat with a smaller data bank size.

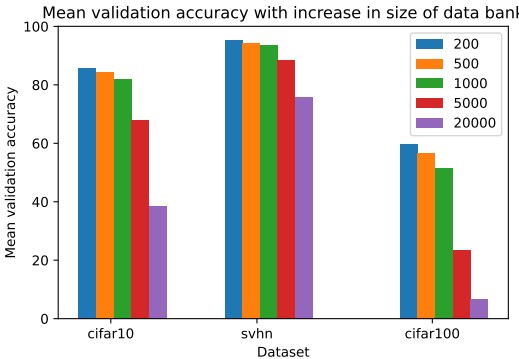

Figure 4: Ablation studies on the size of the data bank and its effect on the model performance (validation accuracy).

## 6   Conclusion

We developed a novel sampling algorithm IMPON that achieves faster convergence with high accuracy using an auxiliary linear model. A key principle that IMPON highlights is using a simple model context to regress the expected loss is an effective strategy to learn the importance weights. In the future, we would like to explore even simpler approximations of IMPON. IMPON's training principles can also be further refined by considering dynamic data bank sizes, ranking losses for supervising the linear model etc.

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
