# OpenReview forum: "IMPON: Efficient IMPortance sampling with ONline regression for rapid neural network training"
_NeurIPS.cc/2022/Workshop/HITY — HITY Workshop NeurIPS 2022_

### Official Review · Reviewer_wHDb · 2022-10-08
**Estimating losses as importance sampling criterion using a linear model**

**Rating:** 0
**Confidence:** 3

**Review:**

The paper presents an importance sampling algorithm that uses a linear model on the input and the neural network's weight norm to estimate the loss during training. The linear model is updated each iteration using the true losses, and its estimated loss values serve as importance scores to draw a batch.

Overall, I like the idea presented in this work. However, I believe that some passages need to be improved to make the manuscript clearer, and the experiments must be improved to be more convincing.

---

Detailed comments:

- Phrasing:
  - I did not understand the full first paragraph of the introduction, specifically how it relates to importance sampling in a broader picture. I believe it could be completely left out, starting with L28.
  - L39-40: I did not understand how computing the per-sample training loss is expensive. Aren't these just the loss values before aggregation?  In fact isn't this stated in L59? It would be good to make the statement more precise.
  - L124: Do you mean "epoch" instead of "iterations". Also applies to ylabels of Figure 2.

- Experiments: I would argue that the plots shown in Figure 2 do not allow for the strong conclusion that "IMPON is able to outperform all the baselines across all the datasets comprehensively":
  - The curves were determined from a single run. Statistics obtained from multiple runs (with varying random seed) would be more convincing.
  - The x-axis shows "iterations" (I think it should be "epochs"). Wouldn't it make more sense to show time here, especially since some algorithms introduce additional computations, and the goal of the paper is to achieve faster training?
  - From the main text it is not clear how much effort was spent to tune the hyperparameters (e.g. learning rate) of both the presented method and the baselines.

---

Miscellaneous comments:

- L33: Formulation "a set criteria" sounds weird to me.
- L35: "incur" → "incurs"
- Double check: "criteria" is the plural of "criterion"
- L60-61: Missing comma or incomplete phrase?
- L72: "is" → "are"
- L75: "choose" → "chooses"

---

### Official Review · Reviewer_6o3n · 2022-10-17

**Rating:** 1
**Confidence:** 4

**Review:**

This work proposes a new method for speeding up neural network training by importance sampling. The paper is clearly written, and the empirical results demonstrate superiority of the method over relevant baselines.

---

### Official Review · Reviewer_4rWQ · 2022-10-20

**Rating:** 1
**Confidence:** 3

**Review:**

**Summary**

The paper presents a method for weighting examples to select in SGD to improve training performance. The method is based on fitting a linear model online to predict the loss of the examples and perform importance sampling instead of re-computing the losses/gradient norm every step or epoch.

**Detailed comments for the authors**

The submission needs additional ablation studies to understand the proposed method

- I command the authors on including the ablation in Appendix A. However, this is a clear sign that something is “wrong” with the motivation of the method as currently presented, or that it needs to be modified to fix this issue. I’d recommend using the same setup on fitting a linear or logistic regression to diagnose the problem, and encourage the authors to further explore this issue before the workshop — this will definitely come up if discussed.
- This workshop emphasizes a fair comparison between methods. It is not clear from the presented experimental results that the improved efficacy of the proposed method over its competitors is due to a better step-size selection. This should be confirmed by a step-size sensitivity plot of grid-search to ensure a fair comparison.
- The inclusion of the norm of the weights of the model is justified as a proxy for its state, but it is poorly motivated. An ablation showing the benefit of this design choice should at least be presented. And the phrasing should refer to `w` as the flattened weight vector rather than matrix, as there is no single “weight matrix” and the weight matrices of a network do not have a well-defined concatenation.

The submission needs to be more precise in its terminology and statements.

- “Training loss or gradient norms […] are generally expensive to compute”; This relies on the unstated problem that sampling requires to have some metric (such as the loss of gradient norm) for all examples. The loss/gradient norm are trivial to compute for the current minibatch, even per-sample in the minibatch. The problem is that we need it before, and previous methods do not actually use any metric at the current state and instead reuse previous data.
- “Alain et al. (2015); Katharopoulos & Fleuret (2018) have proven that choosing examples proportional to the gradient norm can enable faster convergence” (L54). Those references contain empirical evidence that non-uniform sampling helps and provide intuition on how non-uniform sampling leads to faster convergence for neural-networks.
- Line 70; `\epsilon^t` is an undefined quantity.
- “we aim to bridge this gap between computational complexity and[, similar to …], we also use loss values.”
- The plots indicating `Iteration 0` do no contain the values of the model at iteration 0, as they should all start at the same accuracy at initialization.
- In the Weight resets paragraph, the word “stable” is not a good choice in “[The linear model overfitting] resulted in “learning plateaus”. To alleviate this […] results in more stable performance.” A learning plateau is a more stable performance than convergence.

---

### Decision · Program_Chairs · 2022-10-20

Accept